# A Signal Processing Method for Assessing Ankle Torque with a Custom-Made Electronic Dynamometer in Participants Affected by Diabetic Peripheral Neuropathy

**DOI:** 10.3390/s22166310

**Published:** 2022-08-22

**Authors:** Iulia Iovanca Dragoi, Teodor Petrita, Florina Georgeta Popescu, Florin Alexa, Sorin Barac, Frank L. Bowling, Neil D. Reeves, Cosmina Ioana Bondor, Mihai Ionac

**Affiliations:** 1Department of Vascular Surgery and Reconstructive Microsurgery, “Victor Babes” University of Medicine and Pharmacy, 2 Eftimie Murgu Square, 300041 Timisoara, Romania; 2Department of Communications, Politehnica University Timisoara, 2 Vasile Parvan, 300223 Timisoara, Romania; 3Discipline of Occupational Health, “Victor Babes” University of Medicine and Pharmacy, 2 Eftimie Murgu Square, 300041 Timisoara, Romania; 4Department of Surgery & Translational Medicine, Faculty of Medical and Human Sciences, University of Manchester, Oxford Rd., Manchester M13 9PL, UK; 5Research Centre for Musculoskeletal Science & Sports Medicine, Department of Life Sciences, Faculty of Science and Engineering, Manchester Metropolitan University, Oxford Rd., Manchester M1 5GD, UK; 6Institute of Sport, Manchester Metropolitan University, Manchester M1 5GD, UK; 7Department of Medical Informatics and Biostatistics, University of Medicine and Pharmacy “Iuliu Hațieganu”, 8 Victor Babeș, 400000 Cluj-Napoca, Romania

**Keywords:** ankle torque, dynamometer, diabetic peripheral neuropathy, biomedical signal processing, type II Chebyshev filter, feature extraction, level windowing

## Abstract

Portable, custom-made electronic dynamometry for the foot and ankle is a promising assessment method that enables foot and ankle muscle function to be established in healthy participants and those affected by chronic conditions. Diabetic peripheral neuropathy (DPN) can alter foot and ankle muscle function. This study assessed ankle toque in participants with diabetic peripheral neuropathy and healthy participants, with the aim of developing an algorithm for optimizing the precision of data processing and interpretation of the results and to define a reference frame for ankle torque measurement in both healthy participants and those affected by DPN. This paper discloses the software chain and the signal processing methods used for voltage—torque conversion, filtering, offset detection and the muscle effort type identification, which further allowed for a primary statistical report. The full description of the signal processing methods will make our research reproducible. The applied algorithm for signal processing is proposed as a reference frame for ankle torque assessment when using a custom-made electronic dynamometer. While evaluating multiple measurements, our algorithm permits for a more detailed parametrization of the ankle torque results in healthy participants and those affected by DPN.

## 1. Introduction

Electromyography (EMG) and electronic dynamometry are two commonly used methods that assess foot and ankle muscle function in humans. Electronic dynamometry for the assessment of ankle torque is a promising diagnostic method. Enabling measurements of lower limb muscle strength, electronic dynamometry can be used alongside or as an alternative to EMG to provide insight to foot and ankle function. 

Isokinetic muscle testing, despite being regarded as the golden standard assessment for muscle function, involves high costs [1] and special clinicians training.

Custom-made electronic dynamometry due to its portability, manoeuvrability and low cost has gained great interest and shown high reliability and utility as a research and clinical tool [2].

From both EMG and electronic dynamometry, muscle contraction-derived signals are captured, and the arising data require processing and interpretation. 

In comparison with EMG [3] and isokinetic dynamometry [4], for custom-made electronic dynamometry, there is poor information available on the procedures used for processing the signals resulted from muscle efforts. 

Human gait requires foot- and ankle-appropriate muscle performance, which can be altered by various acute and chronic medical conditions such as diabetic peripheral neuropathy (DPN). 

Diabetic peripheral neuropathy affects people’s mobility; therefore, analysing the foot and ankle muscles parameters through any means enables a better understanding of how the foot and ankle function.

Muscle strength is altered by DPN, and studies have analysed muscle function using clinical tests [5], hand-held dynamometry [6] and isometric/isokinetic dynamometry [7], with isometric/isokinetic testing being the preferred method for the assessment of torque and maximal muscle strength. 

Ankle plantar flexors and dorsiflexors are the main groups participating during the stance phase [8]; therefore, measuring these particular muscle strength parameters using precise and reliable methods is relevant for understanding gait performance in people affected by DPN. 

Different papers described custom-made devices for the assessment of ankle torque in participants not affected by DPN. 

In one paper first describing a plate on pivots, a custom-made device was used to assess how the ankle joint position influenced dorsiflexor strength [9]. 

A similar custom-built device with a force cell on a foot plate was used to measure muscle forces acting around the ankle joint [10]. 

Moraux used a custom-made dynamometer to measure ankle torque [11], and the same principles were later used to determine torque around the metatarsal phalangeal joints (MPJ) [12]. Incomplete data from the previous published papers regarding the used custom-made devices’ calibration procedure, voltage measurement solution, the software chain and the signal processing methods applied opened the perspectives for new research. 

By improving the already existing methods, enhanced muscle testing procedures were studied, demonstrating that ankle torque measured by portable, custom-made electronic dynamometry is a reliable [2] and reproducible method [13] in both healthy individuals and those affected by acute conditions.

Introducing custom-made electronic dynamometry in the assessment of ankle torque in chronic conditions, such as DPN, could open the path for innovative methods for strength measurement. 

Through dynamometric measurements of consecutive and repetitive maximal voluntary isometric contractions (MVIC), symmetrically and bilaterally affecting conditions, such as diabetic peripheral neuropathy, could benefit from a precise measurement of some of the foot and ankle muscles parameters. 

Signals derived from human muscle efforts assessed through MVIC in both healthy and affected by DPN participants require that all acquired data are processed for further medical interpretation and analysis. 

Processing the data involves signal conditioning (amplification, filtering, noise suppression, processing of resulted measurement errors), yielding time or frequency graphs, further processed into significant biological parameters, usually by statistical means. Signals derived from muscle contractions on a portable dynamometer were captured, amplified, filtered and analysed as time graphs [2], and the same path was applied for medical interpretation of the resulted data [13]. 

The apparatus used in the present study is a portable, custom-made electronic dynamometer [14], and it represents a replica of the device used by Reeves et al. [10].

This paper details the methodology behind the scientific research. 

The proposed processing algorithm that enables signal filtering and automatic offset detection is able to remove aberrant values from data. 

In this article, to the best of our knowledge, we present for the first time all the steps needed for the collected data processing to obtain relevant measurements results when ankle torque was assessed with a custom-made electronic dynamometer. The aim of the study was to describe the steps for the signal-processing chain and data processing used to assess ankle torque with a custom-made electronic dynamometer in participants with diabetic peripheral neuropathy. The set-up for the voltage signal acquisition, the digital signal processing including scaling, filtering, feature extraction and data processing is described. 

The second objective was to disclose all the encountered situations during measurements, possible errors and the validation considerations of measurements. In order to achieve medical interpretation of data, the obtained results were further used for primary statistical processing. 

This algorithm was intended to be a reference frame in portable, custom-made electronic ankle torque dynamometric assessment for both non-affected and affected-by-DPN participants. 

Analysing foot and ankle muscle strength is essential in the presence of DPN as part of the quantitative assessment. As foot and ankle muscle strength are difficult to precisely measure by manual tests, ankle torque dynamometry could offer reliable and accurate quantitative data for the clinicians to better diagnose strength deficit and further prescribe personalized treatments. 

## 2. Materials and Methods

Ankle torque measurements captured using a portable, custom-made electronic dynamometer in participants affected by DPN and healthy participants were selected for analysis. A total no of 776 measurements from different clinical studies were included (from one study [2], 48 measurements were selected, from which 4 measurements were considered errors; from a second study [13], 96 measurements were selected, from which 3 were considered errors; from a third study [15], 512 measurements were selected; from a fourth study, 120 measurements were selected, from which 1 was considered an error). From the total number of the selected measurements, 48 belonged to participants affected by DPN. Due to the large amount and complexity of the generated data, an automated signal processing approach was applied and tested. The acquisitions resulted data were processed and analysed for later clinical interpretation. All the data belonging to consenting participants were covered by informed and signed consent before the enrolment. All the studies were conducted in accordance with the Declaration of Helsinki and approved by the Ethics Committee of the University of Medicine and Pharmacy “Victor Babes” Timisoara, released and registered under Nr. 50/21.09-14.10.2020.

### 2.1. System Hardware Components

The hardware is briefly described here for reference. A detailed description of the hardware system and its components are presented elsewhere [2,13,15]. 

The measurement system hardware components were: a portable, custom-made electronic dynamometer designed for the measurement of ankle torque [2], manufactured by Research Solutions (Alsager, UK) [14], with included load cell [16], load cell amplifier [14] and oscilloscope [17], connected through wires cable to a personal computer (PC). An image of the measurement system set-up/layout is represented in Figure 1a, while a complete diagram of the measurement system hardware components is represented in Figure 1b.

The measurement device (portable, custom-made electronic dynamometer) consisted of a suspended aluminium pedal and a weight-measuring load cell-CZL-601 [16] with an incorporated strain gauge rated at 100 kg connected as a classical Wheatstone (resistive) bridge. 

The manufacturer recommendations on the calibration procedure for the device used in the current paper were applied prior the measurements. The complete apparatus calibration procedure is described in detail elsewhere [2]. 

The measurement device pedal inclination being changeable by selecting the degree angle using an electronic inclinometer permitted ankle torque to be measurable at different ankle joint angles. Measurements at 0°, +5° and −5° inclination were selected for this study. The measurement device system provided a voltage directly proportional to the torque further converted into force by software means. 

The measurement device design allowed for standardization of ankle torque measurements, the standardization being achieved by placing the participants in a resting seated position having their tested lower limb fixed in place using the apparatus’s rigid fixation straps. 

Participants generated ankle plantar flexion or dorsiflexion contractions, and the obtained signal was readable by the load cell transmitted through a four-wire cable to the load cell amplifier that further converted the Wheatstone bridge imbalance into voltage. Voltage was later evaluated with the oscilloscope connected to the PC. 

The oscilloscope (PicoScope2204A) came with its manufacturer software PicoScope^®^6, freely available on the same producer’s website [18]. 

The oscilloscope software memorizes the whole movement and produces graphs of the recorded torque.

### 2.2. System Software Components

#### 2.2.1. Data Acquisition Software

For all selected measurements, the PicoScope^®^6 software [18] was used according to the manufacturer specifications. PicoScope^®^6 software allowed for the selection of particular parameters depending on the requested type of analysis. 

A complete guide for the selection of configuration parameters for all the selected data is detailed in Figure 2 and was previously used in the same manner [2,13]. 

The format of the measurement data resulted upon acquisition is described in Section 2.3.

#### 2.2.2. Data Processing Software 

In order to process the selected data, various GNU Octave and MATLAB scripts were used. Since the work was completed on diverse computers, not all equipped with a MATLAB license, scripting was written to maintain compatibility between the two suites as much as possible. 

The processing chain was performed mainly in the classical computer algebra system MATLAB [19], but the presented code is compatible with Octave suite [20], which can be regarded as a free version of MATLAB. The summary of processed data was further exported in Microsoft (MS) Excel format [21] for further statistics use. Final statistics data, as previously reported [2,13], were mainly processed from these summary files.

A notable difference of execution speed was observed between the two suites (MATLAB and GNU Octave) regarding the *for* loop, which could be one order of magnitude faster on MATLAB. *For* loop was essential in multiple folder processing, which could lead to long processing times. The loop time for a measurement could be of tenths of seconds, so a large measurement sample processing time could reach 30 min on available computers. Repeated runs performed during the development of the scripts led to a significant time consumption.

In order to assess the results in a simpler manner, the software scripts were developed in a Jupyter notebook [22] installed within an Anaconda environment [23]. 

The configuration of MATLAB and GNU Octave in order to work within Jupyter environment needed some attention [24]. 

The results were presented as graphs made by MATLAB/GNU Octave scripts but also summarized as relevant numbers in an Excel sheet.

### 2.3. Data Processing 

Primary data results after the ankle torque acquisition process were not suitable for medical interpretation in the absence of a specific data processing procedure. For this reason, a processing algorithm followed. The steps for data processing targeted: formatting the acquired data, collection of data and file indexing, analysing the pedal signal, low pass filtering, offset detection and feature data export.

#### 2.3.1. Acquired Data Format

The recorded measurements belonging to a certain collection of measurements were grouped in a folder—with all recordings saved as files with an alpha-numerical code so that participants’ data remained anonymous for the data processor and statistician, respecting thereafter the ethical and data protection requirements. 

The recordings were stored in multiple text files, every text file being a second of memory buffer. For convenience, we worked with 32 buffers (default software choice), so all our measurements so far had a consistent length of 32 s. The number of buffers can be configured to a different number, but most of the presented processing here are not dependent on the buffer choice and should work with other lengths.

Besides of its own format, the software can save all the buffers in separate files (namely 32 files) in a folder of choice. This kind of export is available as common formats such as *.txt*, *.csv* or *.mat* (MATLAB matrix data format). We chose the *.txt* format, since it is the simplest one. In the future it is intended to test other voltage acquisition gear; this will impact mostly on this particular section of the processing chain, the other steps of processing being suitable for other voltage acquisition gear/equipment.

The text files were stored in a folder for each performed measurement. Each text file was ordered by two rows, first being the time moment, and the second being the measured voltage. The time starts for every buffer from zero, so for the time axis, time needs to be appended properly. The file name had the same folder name, but appended with a two-digit number, which points to the buffer succession, this being the feature used in the MATLAB/GNU Octave script in order to properly arrange the data. The first three rows of each file had to be discarded, since they contained the file header.

Reading a measurement folder means taking the data from all files in proper order, concatenating the data and calculating the time axis, yielding into a graph of 32 s (for the 32 buffers recording), which shows the whole time-function of the dynamometer pedal press.

#### 2.3.2. Data Collection and the Index File 

Measurement data were collected from different studies that included participants both affected and not affected by DPN (healthy participants).

All data were obtained from measurement sessions performed in different occasions, in different moments, days or even weeks intervals acquired in Timisoara in the same Physiotherapy Unit between October 2020 and June 2022.

Data from one measurement session must gather all measurement data from all participants that were included in that particular session. 

Further steps required that data from all sessions were concatenated in one folder. 

For collected data belonging to one participant, a unique two-digit code number was released. Some biometric features needed for statistical purposes were kept and attached to the index file. 

The directory name containing the collected data were gathered in an Excel file including measurements in different modes or conditions (plantar/dorsiflexion, left/right foot, ankle angle degree). 

#### 2.3.3. The Pedal Signal Analysis

In case of three MVICs (e.g., during plantar flexion), a typical pedal multiple flexion record looks like a rectangular signal with 3 periods in the ideal case as seen in Figure 3a, having a period close to 10 s. One muscle effort for acclimatisation of participant with the requested type of contraction is represented in Figure 3b, and this is a typical singular contraction/muscle effort graphical representation. A period is defined by the interval between two successive MVICs. The amplitude of the signal stays within 1 ÷ 2 V peak to peak (Vpp) with an offset of few hundred millivolts (mV). The offset is caused by the remanent pedal torque and the foot weight in a relaxed state including the torque derived from the fixation strap.

There were two main problems to solve: the offset measurement and the contraction measurement in respect with the acquired offset. The measurement should yield at least one value, namely the MVIC value.

#### 2.3.4. The Low Pass Filtering 

As one can see, the signal can be noisy (Figure 4a). Since the signal power density is relevant at much lower frequencies (Figure 4b), it can be filtered quite easily. We tested multiple low pass filters in order to find a best fit. Since we expected the shortest pedal action to be about 1 s, in order to enclose the 11th harmonic, the low pass bandwidth should have been at least 12 Hz. On the other side, there may be a hum component, depending on the electric network interference at the place of the measurement. 

We used a low pass filter in order to eliminate the hum noise. The signal can look different depending on the PC power supply. In the case of using a laptop, two different situations may be encountered: PC working on battery supply as seen in Figure 5a, and PC working with electric network supply as seen in Figure 5b.

Since a Chebyshev type II filter experienced rejections in the stopband, we made the decision to make use of it in order to achieve a lowpass characteristic and a network frequency rejection.

A Chebyshev filter is a signal filter in which coefficients are calculated from Chebyshev polynomials, and it is one of the classical filters largely used in digital signal processing. A Chebyshev type I filter has ripple in the passband and a flat stopband, while a type II, or an inverse Chebyshev filter, exhibits ripple in the stopband, having a flat passband [25,26]. In the stopband, the ripple moves towards zero value of transfer function at certain frequencies, yielding to notches into the stopband. 

The Chebyshev filter is cited in various bioelectric signal processing works, mainly for electroencephalogram (EEG) and electrocardiogram (ECG) signals, and some of them use the idea of the inherent notch included in Chebyshev type II filter stopband [27,28]. As it can be observed from previous section in Figure 4b, the pedal press-derived signal exhibits similar bandwidth characteristics as EEG and ECG signals, having relevant energy much under 10 Hz. As it was still in an experimental phase, we maintained the processed signal bandwidth relatively high, until further experiments could be performed with different hardware.

Since electric network frequency (ENF) exhibits a small variation, we chose a higher order (5th) filter in order to tune the second notch into the ENF value, due to the fact that the second notch is larger and can cover a higher variation of rejected frequency. As one can see, although the two Chebyshev filters (type I and type II) were designed with the same parameters, namely 5th order with a cut frequency of 29.3935 Hz, the actual cutting frequency is smaller for the type II filter. The first notch is around 27 Hz, which has no practical meaning for the filtering characteristics, since is not used *per se*. The exact value of the design cutting frequency is chosen in order to match the second notch at the ENF peak; with these values the filter exhibited a minimal attenuation of 78 dB in the 49.5– 50.5 Hz range. The designated cutting frequency appears as a circle in Figure 6.

The other design parameters were the 1 dB ripple in the passband of type I (just for reference, irrelevant in the matter) and the −50 dB stopband for the type II, as it can be easily seen in Figure 6. The actual cut-off frequency of the type II Chebyshev filter is 15 Hz, which yields the signal bandwidth. This value fills the 12 Hz condition, as showed in the previous section.

#### 2.3.5. The Offset Detection

One of the most important steps of the measurements processing is the detection of the pedal offset. This can be completed with the prior knowledge of the contraction type or without. The pedal exhibited an offset voltage itself, due to its own weight. The foot weight due to the participant leg position adds to this value a fixed offset value that cannot be used in order to compensate for the non-contraction part. As a consequence, the offset must be evaluated for every measurement. The evaluation with the oscilloscope consumes time and needs a specialized operator; moreover, the offset data are retained anyway in the recorded waveform, so an automatic detection is very useful.

The offset part is relatively stable, and in case of a correct measurement, it occupies about 50% of the time. The contraction should be the other 50% of the time, but the value is unlikely as stable as the offset value. The offset value can have some drift, which we can hypothesize is due to the movements of the participant, but our experiments showed it as relatively low. A highly variable offset should be subject of measurement rejection. Due to these considerations, we tested the offset detection as the maximum of the measurement histogram, as seen in Figure 7.

#### 2.3.6. Feature Data Export

Once all data were processed, there was a need for the feature extraction in order to use the acquired measurements. Since we worked on cohorts, and the features were interesting at this time for statistics, it was useful to pass to the statistician the raw conclusions of the measurements. One of the simplest forms it was in was a classical spreadsheet software, which allowed everyone involved in the research access.

Since both MATLAB and GNU Octave can easily export in MS Excel format, this was our obvious choice. We describe this in more detail in previous papers [2,13]

### 2.4. Primary Mathematical Calculus

Feature extraction implies primary statistical calculations from measured samples such as average dispersion or peak. The first feature extracted was the offset estimate, which was extensively treated. Most important, once the offset is known, it is the maximum or minimum value of the measured voltage depending on the contraction type. The difference between this value and the estimated offset yields the MVIC value. From primary extracted features there can be derived various statistical parameters. A description of the primary mathematical calculus for voltage to torque conversion was, in detail, presented elsewhere [2]. Other possible parameters that can be derived from the measurements in order to be further used for medical interpretation are: the variation of force during MVIC, the variation of force between successive MVIC during one acquisition, force staidness. 

## 3. Results

### 3.1. The Set-Up for the Voltage Signal Acquisition

In order to obtain the data acquisition, we used the hardware set-up. The set-up for the voltage signal acquisition was already described in Section 2.1, Figure 2.

### 3.2. The Digital Signal Processing

#### 3.2.1. Data Collection and the Index File 

The resulted directory name containing the collected data gathered in an Excel file including measurements in different modes or conditions (plantar/dorsiflexion, left/right foot, ankle angle degree) are presented in Table 1. 

Flags were attached for measurement identification. For each measurement, a plantar flexion or dorsiflexion flag and a left or right foot flag were mandatorily added. Depending on the data provenience and the study requirements, there were various flags attached to a measurement (e.g., *pre-* or *post-*acute condition status, or the rest time before previous contraction). All these flags were mirrored in the folder title (e.g., ‘*20210909-0001 Sub 01,LFT,+5,PFlex*’) in order to determine what the measurement folder represent. After multiple measurements the coding became complicated and non-rigorous, a small database was initiated in a MS Excel file. This MS Excel database held the main information on the measurement type, linking each participant to their measurements. 

Since the participant data are further needed for statistical purpose, another MS Excel file containing participants’ data was elaborated, having their unique identifier alpha-numerical codes listed in relation to participant biometric data. The same identifier was used for each measurement involving the corresponding participant. The statistician analysed the measurement results and corelated them with the biometric data using specific statistical tools.

The algorithm for feature extraction is described in Figure 8.

For every measurement folder, there was a validation process. If the measurement passed the validation process, a feature extraction followed. Most of this paper content is concerned with the algorithms behind measurement validation and features extraction. In a measurement cohort there are a number of N measurements performed. Not every measurement will pass the validation test. Examples of measurements that failed the validation test due to errors encountered during the measurements have been previously showed [2,13].

After multiple measurements with encountered errors were identified, we concluded that an index file comprising the relevant measurement description might be helpful; nevertheless, most of the scripts we ran used the detection of the relevant parameters from the measurement file name. In order to be computer-detectable, certain keywords were used for the detection of the measurement type as detailed in Table 1. 

The keywords were chosen by the main researcher to ensure a later identification of a particular measurement.

As an example, the folder name ‘*20210909-0001 Sub 01,LFT,+5,PFlex*’ means this measurement belongs to the subject (participant) 01, and it is a plantar flexion made with the left foot with the initial pedal angle of +5°. The first digits are generated by the oscilloscope software and represent the date of the measurement and the current measurement order number in the respective date.

A database index Excel file was conceived for measurements, linking the unique participant identifier with its biometric data (sex, age, weight, height, foot length) and other relevant information (e.g., COVID or diabetes status or presence of diabetes-related neuropathy). 

This database concerns the final statistical evaluation and not the measurement processing chain. 

#### 3.2.2. The Low Pass Filtering 

We took the decision to use a low pass filter in order to eliminate the undesired interferences. The filter main function was to eliminate hum noise and other eventual interference signals. 

The proposed solution (Chebyshev low pass filter type II) was tested for stability between sampling frequencies of 4000 Hz and 50,000 Hz. If the sampling frequency was lower than 4000 Hz, the filter needed to be redesigned, since the rejection frequency shifted downwards and above 50,000 Hz. Therefore, another solution was needed since the filter exhibited numerical instability. Our exact sampling frequency was 6103.5469 Hz, which was the sampling frequency of our oscilloscope at the used configuration. The design command worked equally in MATLAB and GNU Octave, and it fit any sampling frequency in the stability interval. The filter coefficients were determined with the command:*[b2,a2] = cheby2(5,50,29.3935/(fs/2)),*
where *fs* is the actual sampling frequency of the signal in Hz. In case of GNU Octave, the loading of the *signal* package (as *pkg load signal*) [29] was necessary in order to design and use digital signal filters; in case of MATLAB, the *Signal Processing Toolbox* was needed [30]. 

After applying the filtering, the measurement result can be seen in Figure 9. 

#### 3.2.3. The Offset Detection

Our best hypothesis was that a maximum of a probability density will indicate the most probable offset value. The trials revealed that during some of the measurements, the foot was unstable on the pedal or the instructions were not followed accordingly, so the maximum of the probability density was not in the pedal offset. A windowed approach considerably reduced the false offset detection, so our final approach was to window the probability density function (as histogram) and to detect the histogram maximum as the most probable offset as seen in Figure 10.

In Figure 10, one can see a distribution of offsets for a cohort of 93 measurements. As revealed later, not all offsets are in the histogram maximum. Figure 7 shows an ideal case, where the histogram maximum is perfectly clear. A slightly unstable foot or a little bit of balancing led to an offset displacement, as it can be seen in Figure 11, where the maximum of the histogram at the pedal press was more stable than the maximum of the offset position.

While testing our method, we observed that some of the histograms (Figure 11) were weighted towards the contraction maximum, though remaining valid. Due to this, we took improved detection by picking the maximum from a windowed histogram. Establishing limits for the offset variation led to an improved detection, recovering thereafter some of the otherwise missed measurements.

In Figure 12, one can see a representation of all histograms (or estimated probability density functions pdf) of measured data in the sample (cohort). In Figure 12a, one can easily observe the positioning of the maximum of the histogram, our presumptive offset (around 0.5 V). In Figure 12b, the logarithm of the histograms shows a few local maximums on most of the measurements. 

Some of the estimated maximum pdfs were not in the right position for the offset to be estimated. These situations are shown in Figure 13. A countermeasure used for fake offset detection is offset windowing. By making multiple observations about pedal offset range, we decided to limit the offset detection window to a voltage window around expected mean pedal offset. This can be seemed in Figure 13, where in upper side, the red line represents the detected histogram position over the entire range, represented here as the histogram interval on the Oy axis, while on the Ox is the index of the measurement. On the lower side, there is histogram windowing, and thus, some of the values are eliminated. 

One can see that there are fewer eccentric values out from the mean line, and some of the histogram maximum positions were changed. By correctly choosing the windowing limits, the error rates improved with one magnitude order. Practically, in one of our analysed cohorts, the 93 measurements, the fake offset detection was eliminated for all the valid measurements. 

#### 3.2.4. Data Processing

Having the offset calculated, the most valuable data from a valid measurement were the MVICs. These were extracted as the difference between maximum voltage value and the offset value for the plantar flexion and the difference between offset value and the minimum voltage value from the dorsiflexion. The detected peaks are represented with round circles in Figure 14.

The detected offset was exported into an Excel file, along with participant-relevant parameters (sex, age, health status, etc.), in order to perform statistical processing.

### 3.3. Errors 

The second objective of our work was to disclose all the encountered situations during measurements, possible resulted errors and steps required for the measurement validation that preceded the statistical and medical interpretation.

Errors resulted during measurements. There were two main types of errors: non-human-related errors (instrumental and method errors) and human errors. 

Non-human-related errors were automatically detected by the proposed algorithm. Non-human-related errors affected the method accuracy. The proposed method can be affected by some particular artifacts induced by the presence of unwanted effects related to the participant interaction with the equipment and by the intrinsic errors introduced by the instrumentation itself and by the measurement procedure. 

The measurement method exhibits a certain degree of error, as does any measurement method. 

Instrumental errors were due to the measurement errors and the limit induced by the used oscilloscope precision, the oscilloscope limited dynamic range, hardware chain, software chain and the pedal sensor errors. 

The pedal errors were discarded since the used apparatus was an experimental prototype. The dynamometer was calibrated with weights, and thereafter, the domain was rescaled according to the requirements of the proposed experiment.

The most problematic error source related to the hardware chain was the oscilloscope resolution. Since it had only 8 bits for a symmetric voltage full scale of ±2 V, the voltage quanta would be 15.625 mV, which would lead to an approx. 0.5 Nm torque error, an overwhelming value among the other sources of errors. In the absence of a more in-depth analysis, which is to follow in further studies, our conservative estimation of an error margin was 0.7 Nm ±5%.

The main human-related errors were due to the participant or the tester. Participant-related errors were caused by indiscipline and possible adverse reactions encountered during measurements (pain, fatigue, emotional reactions, etc.). Tester command errors were mainly due to improper commands, lack of concentration/attention, inappropriate timing of commands, etc. Errors automatically detected by the algorithm were mainly due to improper number of contractions. Inconstant pedal press during contractions and offset instability might result. Such situations require special attention from the operator. As some automatically detected possible errors could have been normal situations encountered in the case of those participants affected by medical conditions, such as diabetic-related peripheral neuropathy, an operator interpretation was mandatory for the validation of measurements. 

Errors that the algorithm could not detect needed a tester/operator special attention. 

#### 3.3.1. Participant-Related Errors

Errors derived from participant indiscipline were encountered during the measurement sessions and did not pass validation procedure. Such indiscipline errors may have been due to either undesired body movements during the muscle efforts execution, as seen in Figure 15a,b; insufficient number of requested contractions, as seen in Figure 16; wrong direction of contractions; insufficient maintenance of contractions, as seen in Figure 17; delay in contracting the muscle on tester command, as seen in Figure 18a,b; etc. Other encountered participant-related errors showed offset unsteadiness due to the inability of the participant to control the relaxation period as seen in Figure 19b; unsteadiness of offset during breaks between two MVICs, as seen in Figure 19c; and inability to maintain the same level of force during MVIC, as seen in Figure 19d. An example of another invalid measurement due to participant inability to complete a maximal muscle effort during MVIC as per tester command is shown in Figure 20.

Other participant-derived errors were encountered due to adverse side-effects such as fatigue, as seen in Figure 21a, or any other emotional reactions such as pain/discomfort, as seen in Figure 21b, or tremor, as seen in Figure 21c.

Some considerations needed to be made regarding the time graphs that represented a possible normal behaviour of signal during contractions despite looking like an error. 

In the case of healthy participants, some signal behaviours were encountered, as seen in Figure 22a–d. Such examples should be not confounded with errors.

Situations such as those the time graphs show in Figure 19 and the situations seen in Figure 23, could be seen as normal in pathological situations and should not be confounded with participant-related errors. 

Examples of fatigue or tremor might need some attention when the data appertain to participants affected by medical conditions such as DPN. As such time graphs might look like errors but could be the result of the natural behaviour in the presence of the condition, more attention is required. 

#### 3.3.2. Tester Command-Related Errors

An example of invalid measurement due to insufficient number of MVICs might result from either participant indiscipline or from the tester command. Such an example was already described in Figure 16 and are also represented in Figure 24.

#### 3.3.3. Automatically Detected Errors

Errors due to wrong editing of keywords (see below Figure 25 which represents a dorsi with a keyword edited as a plantar) might appear. These errors are mainly due to the operator but also due to the participant. One particular example is that while the operator requested a plantar flexion, the participant executed a dorsiflexion. Such errors could be avoided by special attention of the operator while evaluating the correct execution of the command on the PC screen. 

Automatically detected errors are mainly due to technical issues as seen in Figure 26a,b, or participant-derived errors that cannot permit automatically detection, as seen in Figure 27.

The impossibility to make the automation detection of the offset level could be also due to the participant inability to coordinate muscle efforts and relaxation periods. This further does not allow an automatically detection of the offset level, as seen in Figure 27. 

### 3.4. Resulted Parameters Derived from Measurements

Measurement-derived parameters resulted and were considered for further medical interpretation.

Peak voltage/torque were automatically calculated for each acquisition.

Offset detection and were automatically calculated, and offset variations were analysed. Two situations were encountered: offset steadiness, as seen in Figure 19a, and offset unsteadiness, as seen in Figure 28.

When offset unsteadiness was encountered, different possible situations were considered, and it was either due to muscle fatigue, as seen in Figure 28, or due to the inability of the participant to maintain constant offset level during relaxation period (Figure 19b) or before/after contractions (Figure 19c). 

In the same manner, the variation of force during one MVIC, the variation of force between successive MVICs during one acquisition and force steadiness resulted after the primary mathematical calculus. The clinical significance of our study results is of great importance when muscle strength is assessed in participants affected by DPN. The interpretation of the resulted possible encountered errors could help in establishing the differences between what is normal muscle behaviour and real errors while muscle derived signals are being captured by electronic dynamometry in the presence of DPN. Precise discrimination between errors and normal behaviour in the presence of DPN is mandatory for the diagnostic, risk assessment and treatment strategies. 

## 4. Discussion

As there is no widely accepted method for measuring foot and ankle strength in healthy and affected-by-DPN participants, we appreciate that a custom-made electronic dynamometer offers great potential to become the measurement method of choice because of being non-invasive and having lower costs, a good reliability and reproducibility. Portable, custom-made electronic dynamometry can become a valuable tool when the measurement protocols, software chain and the signal processing methods used are rigorously applied when assessing ankle torque in both healthy and affected-by-DPN participants. 

Describing the algorithm for signal processing used to assess ankle torque with a custom-made electronic dynamometer in participants affected and non-affected by diabetic peripheral neuropathy was one of the study aims. 

The hardware set-up, the set-up for the voltage signal acquisition and the digital signal processing including scaling, filtering, feature extraction and data processing (export and basic mathematical calculus) were the main steps of the used algorithm. 

There are few data available on the previous used custom-made devices with regards to the apparatus construction, calibration and functioning. Previously described custom-made dynamometers used a foot plate with one or two load cells (strain-gauge mounted on the apparatus tongue) [9,10,31,32,33,34,35,36,37,38,39].

In previous papers, we fully detailed the custom-made apparatus used, the construction, calibration, functioning parameters and measurement protocol [2,13] when ankle torque was captured, which was conducted using the same custom-made device used in this study.

Other torque measurements using electrical muscle stimulation previously used similar custom-made devices [40,41,42].

While Marsh [9] recorded only active torque measurements, considering cancelation of the passive torque resetting the DC level of the recording system, we used automatically offset detection during each acquisition. 

Marsh reported using alternative current (AC)-provoked contractions instead of volitional MVIC [9]. 

Force was measured during an isometric voluntary contraction using a pre-calibrated force transducer located under the footplate [10]. 

For torque conversion, Marsh multiplied each weight by the distance between the point of weight application and the axis of rotation, while we used automatic calculation of torque using the apparatus pivotal point position, as previously described [2].

Previous studies that used similar custom-made devices did not report any data or not enough data on the software chain and the signal processing methods used for ankle torque data processing [9,10,11,12]. 

Despite the similar signal processing chain, including low pass filtering, no time recordings were published as time graphs in previous papers [11]. Goldmann used a PC digitizer card (NI 6024E, 12-bit ADC—manufactured by National Instruments, Austin, TX, USA), without published time graphs [12]. 

Marsh briefly mentioned the use of an unspecified type of oscilloscope. The resonant frequency of the system used by Marsh, while the participant foot was in position, was 80 Hz (120 Hz unloaded) [9]. 

Marsh used an oscilloscope (Hewlett-Packard model 141B) for displaying the EMG activity later rectified, averaged by the use of a signal analyser (Hewlett-Packard 5480B) and computed by a programmable desktop calculator (Hewlett-Packard model 9810A).

We used an oscilloscope [17], connected through wires cable to a personal computer (PC).

The steps for the signal processing used to assess ankle torque with a custom-made electronic dynamometer were fully described. As per our knowledge, previous studies did not report the data on the processing chain used when a custom-made electronic dynamometer was utilized for ankle torque measurement.

Moreover, previous published papers that used custom-made dynamometry presented single data points rather than time graphs. Marsh showed only singular contraction time graphs and that no particular signal processing data were reported for dynamometric measurements in voluntary and/or EMG stimulated contractions [9]. 

Time graphs are graphic representations in time of voltage or torque during single/multiple contractions. Our measurements only considered MVIC [2,13].

Multiple contractions (three MVICs) for the determination of average torque/peak torque in order to compensate any variations during acquisitions were used. The representations of data using time graphs were performed for a better interpretation of results (type of muscle efforts, contraction direction, possible encountered errors). Time graphs generated from data reveal main parameters of MVIC. Together with the medical information gathered from the participants files, a further detection of the differences between healthy and affected-by-DPN participants can be obtained and used for medical interpretation. 

Data processing normally is undertaken by a data processor, who might not be the same person who conducted the clinical measurements [2]. 

For this reason, in the data collection and the index file, the main parameters for data processing were established. This step was essential for the recorded files to be sent for simple voltage–torque conversion, signal processing chain preparation, signal filtering, offset detection, contraction type detection and MVIC measurements, which further allowed for a primary statistical report. 

Offset detection needs consideration for each individual measurement. A highly variable offset should be generally subject of measurement rejection. In case of measurements belonging to participants affected by chronic conditions such as diabetic-related neuropathy, the variation of the offset level should be considered as a possible specific situation in accordance with the pathology and should be carefully considered for analysis and not automatically rejected. 

The advantages of using a custom-made electronic dynamometer have been already demonstrated. Beside its reliability [2] and reproducibility [13] when used for the assessment of foot and ankle muscle strength in both healthy participants and participants affected by acute condition, another advantage of our method is that we fully disclosed the software behind the ankle torque measurements and the automatization of certain processes. This particularity permitted that the method already tested on healthy and affected-by-acute-conditions participants be transferred for the assessment of participants affected by chronic conditions like DPN. 

One of the only disadvantages encountered was the long data processing time. For this reason, a solution for a more efficient processing time could be developed in the future.

The advantage of signal processing approach resides in the fact that, while evaluating multiple measurements (in both non-affected participants and those affected by diabetic peripheral neuropathy), it permitted for a more detailed parametrization of the ankle torque results, hence showing premises for a better precision and a more relevant data interpretation of the measurement results than a simple maximal voluntary isometric contraction assessment. While the software is still in early development, the description of the signal processing methods makes our research fully reproducible. During measurements, noise can occur due to electronic components that can alter the whole frequency range. High-frequency electromagnetic radiation may result from WLAN, TV or any other electronic components. 

Our non-invasive approach uses signal frequencies components for analysis of the results. In order to eliminate hum noise and other interference signals, we used a Chebyshev filter. 

Moraux used an analogic low pass filter with a cut-off frequency of 10 Hz [11]. 

Artifacts might derive from movement. In biological-derived signals, low-frequency artifacts such as movement artifacts, appear predominantly in the range of 0–20 Hz [43,44]. 

In our study, movement-derived artefacts were related to participants’ spontaneous body movements during measurements, as seen in Figure 15a,b. Some of the participant-derived movements seen as errors on the time graphs were encountered due to possible adverse side-effects such as fatigue, as seen in Figure 21a and Figure 22b, or any other emotional reactions such as discomfort/pain, as seen in Figure 21b. 

Similarities and differences between measurements belonging to DPN and healthy participants were seen. 

Examples of fatigue might need some attention when the data appertain to participants affected by medical conditions such as DPN. As such a time graph might look like it contains errors but could be the result of the natural behaviour in the presence of the condition, more attention is required when graphs are sent for medical interpretation. Operator intervention is required for the accurate assessment of any participant movements while testing.

The second aim was to disclose all the encountered situations during measurements, possible errors and the validation of measurements.

Encountered errors were mainly due to participant and tester. Other errors were considered technical errors. 

Situations such as the time graphs seen in Figure 19 could be seen as normal in pathological situations and should not be confounded with participant-related errors. 

Examples of fatigue or tremor might need some attention when the data appertain to participants affected by medical conditions such as DPN. As such a time graph might look like it contains errors but could be the result of the natural behaviour in the presence of the condition, more attention is required. In case of healthy participants, some signal behaviours were encountered, as seen in Figure 22a–c. Such examples should be not confounded with errors.

After achieving the second aim of the study, the objective was to evaluate the applicability of the same measurement protocol and data processing algorithm when evaluating resulted data belonging to participants affected by diabetic peripheral neuropathy. 

This work accomplished fully describing the signal processing algorithm of ankle toque measurements in order to obtain primary data for statistical and medical interpretation of the results.

This study did not resolve some encountered problems derived from resulted errors, either automatically or operator-detected.

The limitation of our study was the small sample size for the measurements belonging to participants affected by DPN when compared to the sample size of the measurements belonging to healthy participants. For more statistical representative results, a bigger sample is needed. 

Future research should consider for the measurement of ankle torque the use of reliable and reproducible custom-made electronic devices like the apparatus used in our case.

Chronic conditions such as diabetic peripheral neuropathy (DPN) alter foot and ankle muscle function and could benefit of more accurate ankle torque measurement to better understand how this particular condition affects muscle parameters and performance. Future work should address analysing the force parameters such as force steadiness and variation of force during MVIC in participants affected by DPN when compared to healthy individuals. 

Analysing foot and ankle muscle strength is essential in presence of DPN and is part of the clinical assessment. As foot and ankle muscle strength can reach high values for the strength to be manually tested, introducing ankle torque dynamometry could bring more reliable and accurate results for the clinicians to better diagnose strength deficit. Analysing muscle strength requires that all muscle performance parameters are checked. Strength is only one muscle performance parameter frequently assessed through the MVIC value. Muscle strength is individual-dependent. No universal references for normal muscle strength have been defined. For this reason, analysing other muscle parameters such as force variation between repetitive contractions and force steadiness could complete the quantitative data. Such quantitative data could offer more precise information for the risk analysis and personalized treatment.

Future research should consider the analysis of MVIC main parameters such as variation of force during MVIC, variation of force between successive MVICs during one acquisition and force variation in time and force steadiness during MVIC. 

More research is needed for the interpretation of the resulted data for clinical interpretation in presence of DPN. The same resulted data could be used in the future for the quantitative analysis of the treatment evolution and outcomes.

## 5. Conclusions

When the foot muscle function is the main measurement goal, dynamometrical captured muscle efforts-derived signals from both healthy individuals and individuals affected by conditions need a better understanding. When portable, custom-made electronic dynamometry is used with the intent of measuring ankle torque in participants affected by DPN, signal processing is required, and signal characteristics as seen on the time graphs should be carefully interpreted from a medical perspective. 

Not all distorted oscillograms are errors. Captures appearing as possible errors need comprehensive analysis when the measurements are related to chronic conditions as DPN. When DPN is clinically present, distorted signal detected by time graph interpretation may not represent errors, but the natural aspect of signal in presence of the pathology.

Analysing foot and ankle muscle strength through ankle torque dynamometry is essential in the presence of DPN, and quantitative resulted data could offer more precise information for the risk analysis and personalized treatment.

Future research should consider the analysis of the differences obtained on the time graphs between healthy and affected-by-DPN participants. Such differences between the resulted parameters could be explained by the natural course of the pathology and further be used to screen or detect pathology. 

Analysing in detail the differences between spectral components belonging to healthy and affected-by-DPN participants could open new perspectives in DPN diagnostics. Specimens of discarded measurements could be useful as reference for future studies and trials.

New research should consider the analysis of false errors as possibly being the first signs of DPN in the early stage of the condition.

Electronic custom-made dynamometry that includes a precise working algorithm can become a valuable tool not only to accurately measure ankle torque but also in the rehabilitation pathways and progression and treatment outcomes in participants affected by DPN. 

## Figures and Tables

**Figure 1 sensors-22-06310-f001:**
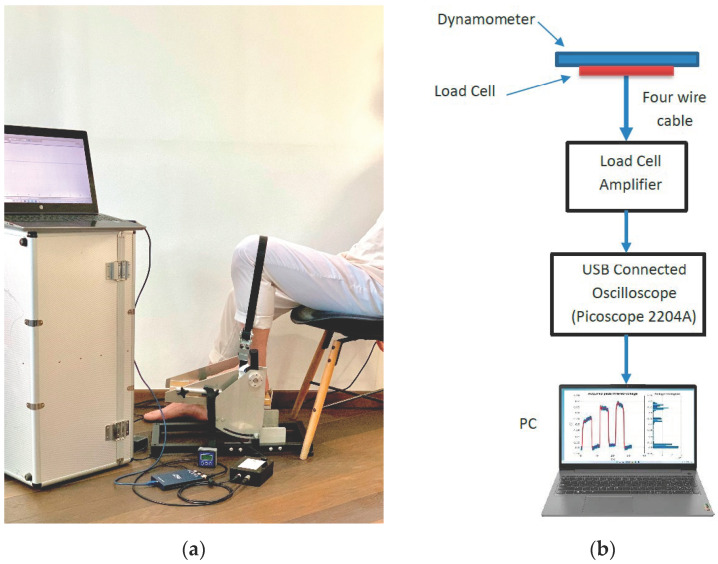
Representation of the measurement system set-up; (**a**) measurement system layout comprising of custom-made electronic dynamometer with incorporated load cell and hardware components: oscilloscope, load cell amplifier, electronic inclinometer, connective wires and personal computer (PC). The installation of the whole measurement system including participant chair demonstrates the reduced required space (approx. 2 m^2^); (**b**) a complete diagram of the measurement system components representing the hardware elements used as graphics: portable, custom-made electronic dynamometer for ankle torque assessment with incorporated load cell, load cell amplifier, oscilloscope, connecting wires, personal computer.

**Figure 2 sensors-22-06310-f002:**
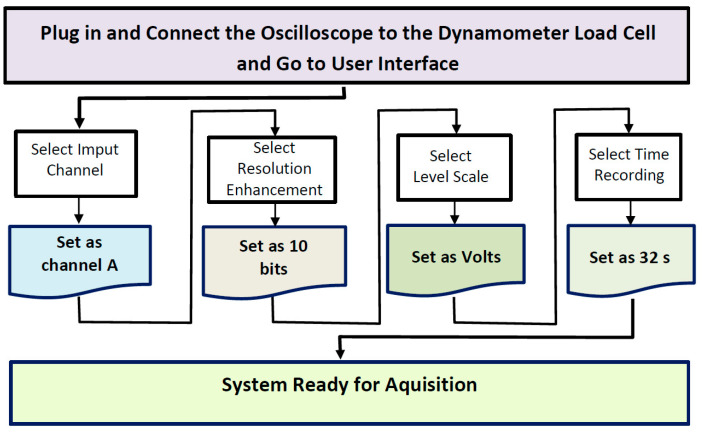
Oscilloscope graphic user interface with complete parameters selection guide used for the data selected for analysis.

**Figure 3 sensors-22-06310-f003:**
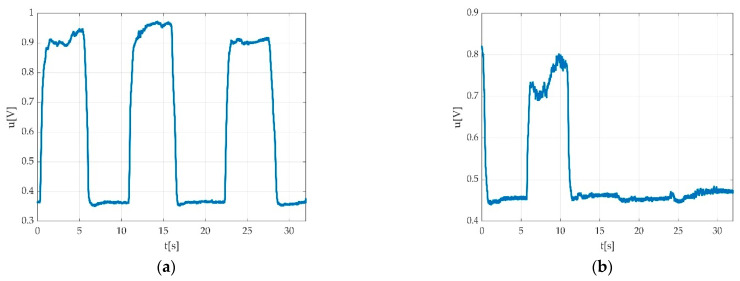
Example of signal acquisition during: (**a**) three consecutive MVICs during plantar flexion with obtained offset (red intermittent line) during acquisition/familiarization with measurement procedure; maximal value of voltage (peak voltage); (**b**) one muscle effort for acclimatisation of participant with the requested type of contraction.

**Figure 4 sensors-22-06310-f004:**
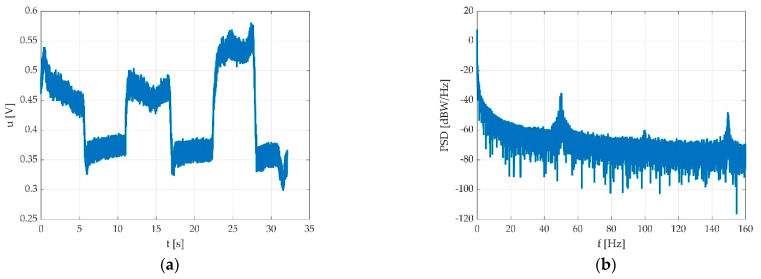
Typical pedal signal due to multiple consecutive flexions: (**a**) time graph with hum noise; (**b**) periodogram of power spectrum density (PSD).

**Figure 5 sensors-22-06310-f005:**
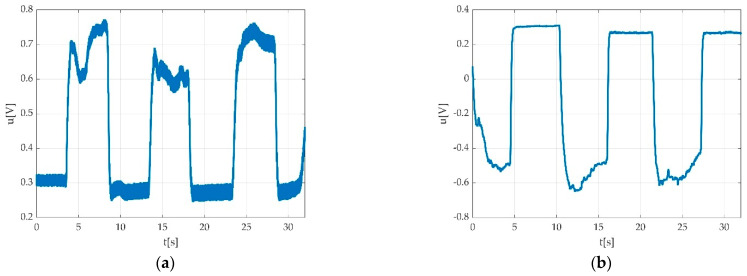
Example of the power supply network influence on the recorded signal: (**a**) PC running on battery; (**b**) PC running while connected to power supply network.

**Figure 6 sensors-22-06310-f006:**
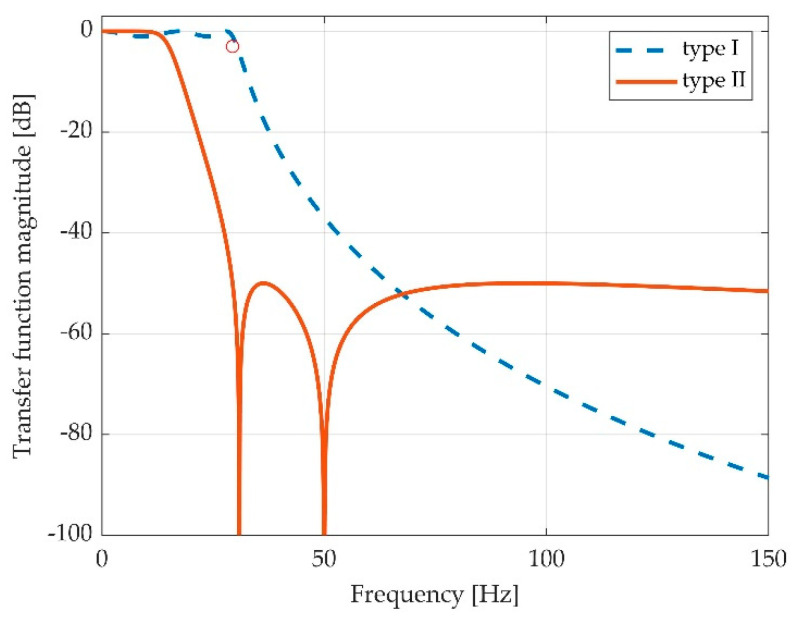
The low pass Chebyshev filter type II frequency response (in orange line) versus same filter of type I frequency response (in blue interrupted line).

**Figure 7 sensors-22-06310-f007:**
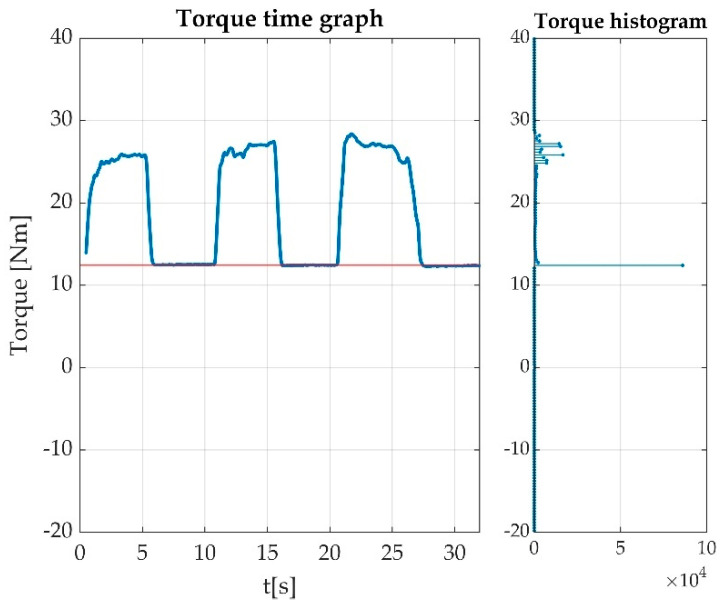
A successful offset detection as the histogram maximum. The histogram is represented in the right part of the time graph. The maximum value on the histogram represents the measurement offset value marked with a red line on the time graph.

**Figure 8 sensors-22-06310-f008:**
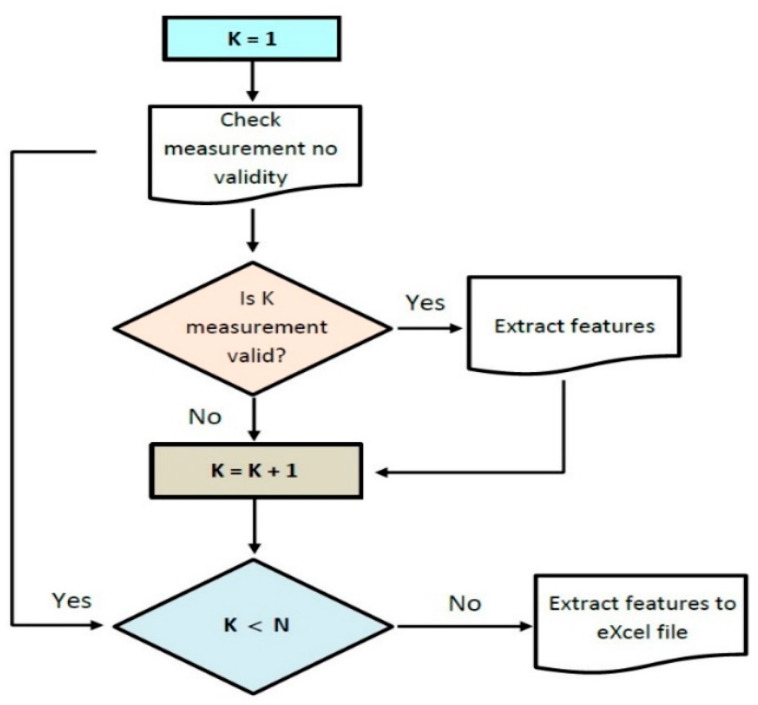
The algorithm for feature extraction from collected data belonging to all participants from a single study. K represents the current position of data; N represents the total number of measurements included; K = K + 1 represents the indexing of current position for all records to be passed through.

**Figure 9 sensors-22-06310-f009:**
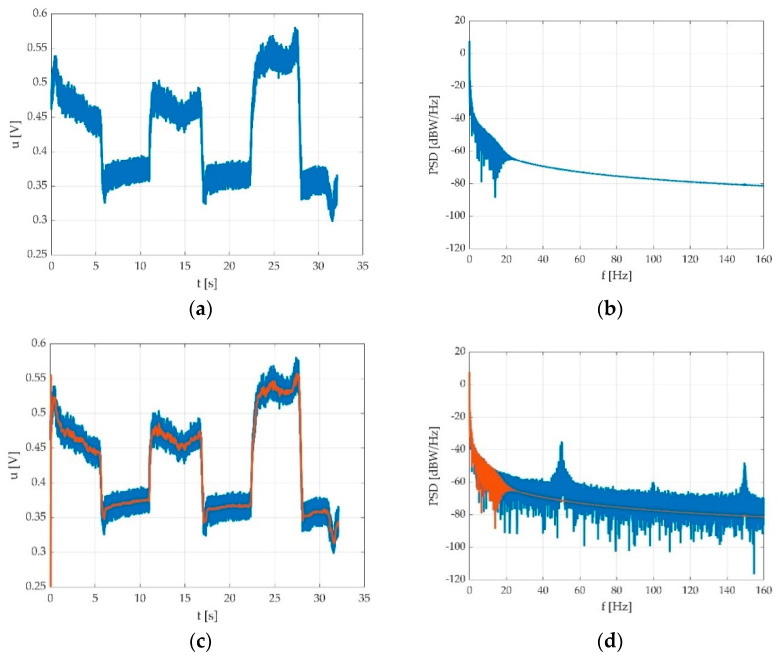
Results after applying the low pass filtering: (**a**) signal after filtering; (**b**) signal spectrum after filtering; (**c**) overlap between original (blue) and filtered (red) signal; (**d**) overlap between original spectrum (blue) and spectrum after filtering (red).

**Figure 10 sensors-22-06310-f010:**
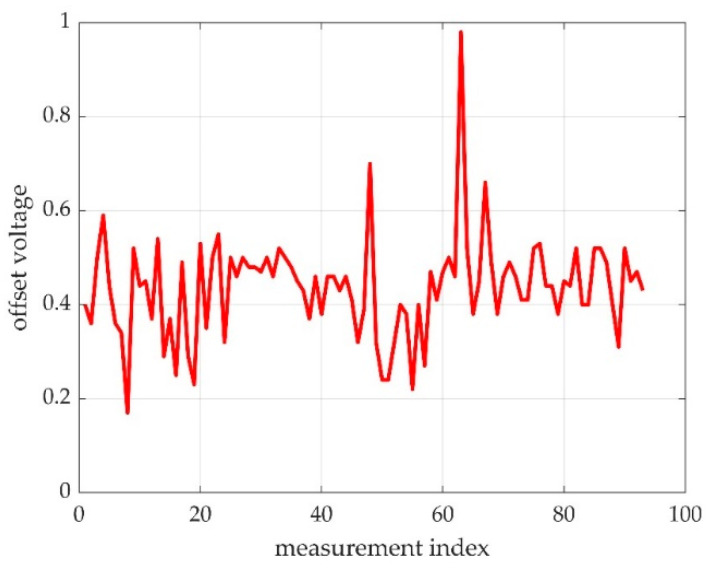
Detected offsets in a set of 93 measurements. About 4% are erroneous detection; for the point approaching 1V level, the error is obvious.

**Figure 11 sensors-22-06310-f011:**
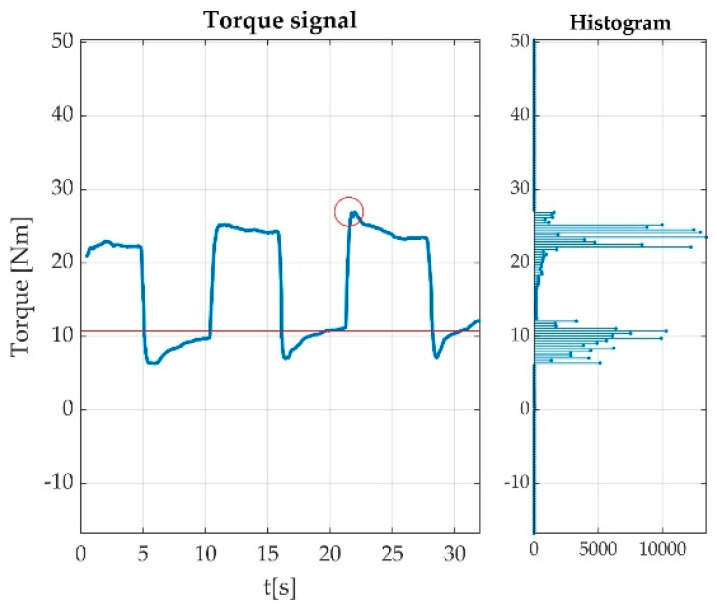
An example of erroneous detection; the maximum of histogram appears at the flexion level, rather than at the offset level; the red circle represents the maximum of histogram during plantar flexion, which does not represent the offset level, instead represented by the red line.

**Figure 12 sensors-22-06310-f012:**
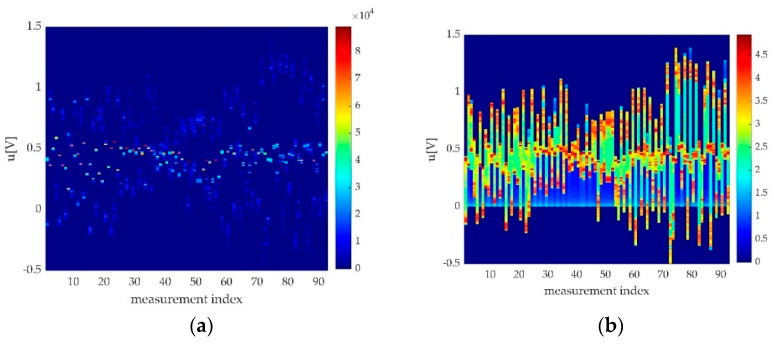
Histograms of measurement cohorts seemed from above colour-coded (histogram of histograms): (**a**) absolute value; (**b**) logarithmic representation.

**Figure 13 sensors-22-06310-f013:**
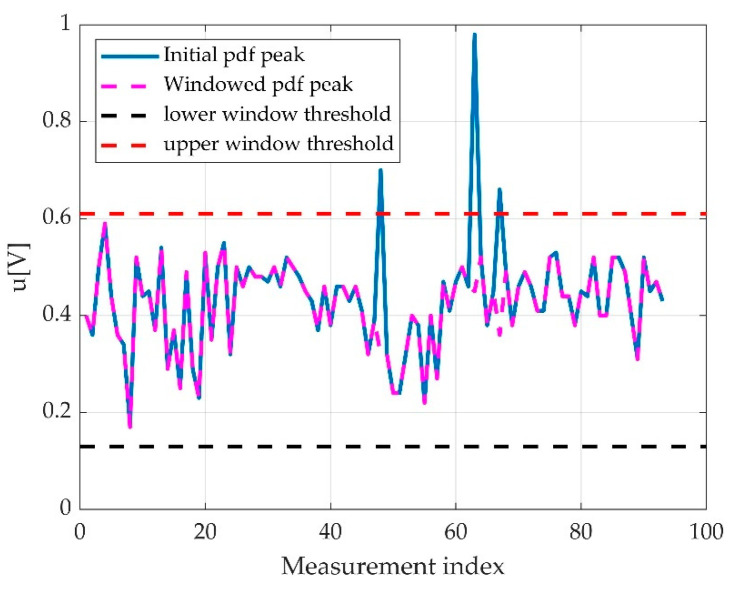
An unwindowed versus a windowed offset detection. It can be observed that in the windowed version, the variance is smaller.

**Figure 14 sensors-22-06310-f014:**
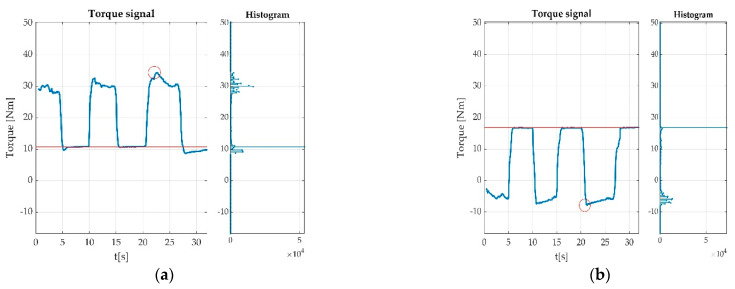
Offset and Peak value detection during acquisitions: (**a**) MVIC during plantar flexion; (**b**) MVIC during dorsiflexion.

**Figure 15 sensors-22-06310-f015:**
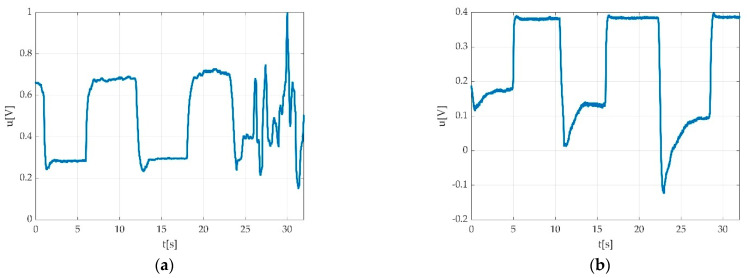
Improper recording due to participant indiscipline: (**a**) participant moved/shifted the trunk during plantar flexion measurement, altering the measurement-resulted signal; (**b**) participant shifted the trunk backwards while tensioning the strap during dorsiflexion, resulting in a sudden increment of voltage value.

**Figure 16 sensors-22-06310-f016:**
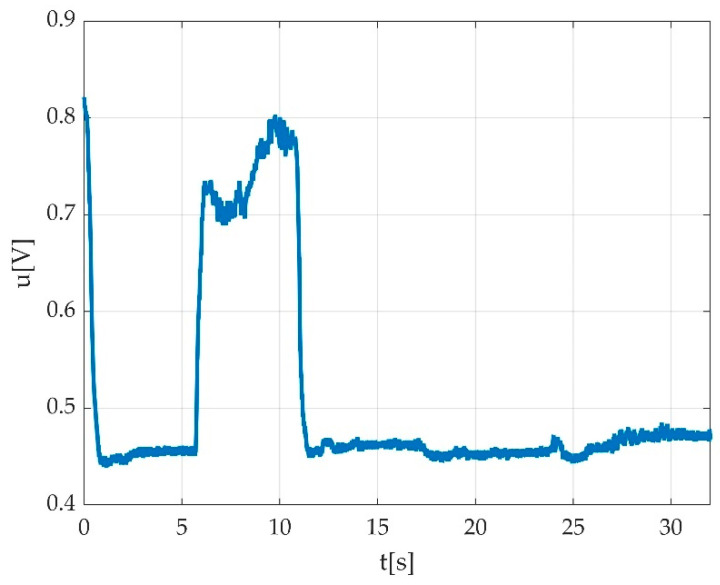
Example of an invalid measurement due to insufficient number of requested MVICs.

**Figure 17 sensors-22-06310-f017:**
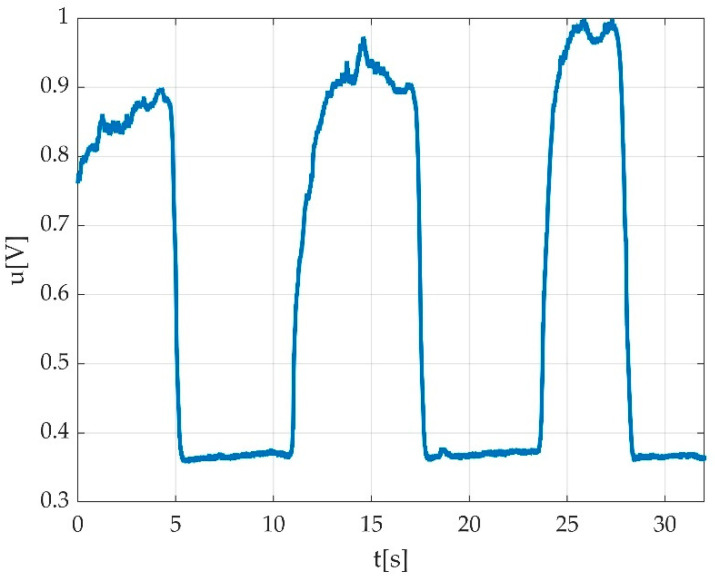
Example of measurement with non-uniform time interval during MVIC, the longer contraction time (third MVIC).

**Figure 18 sensors-22-06310-f018:**
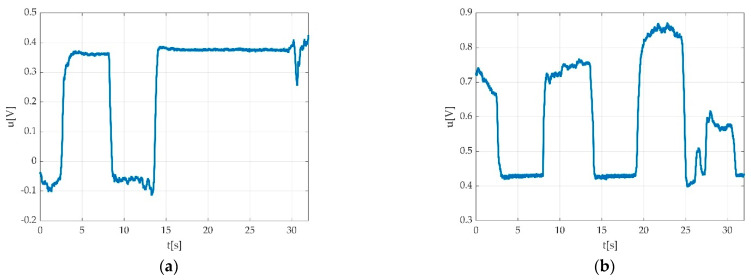
Example of errors due participant delay in contractions the muscle on tester command: (**a**) during dorsiflexion; (**b**) during plantar flexion.

**Figure 19 sensors-22-06310-f019:**
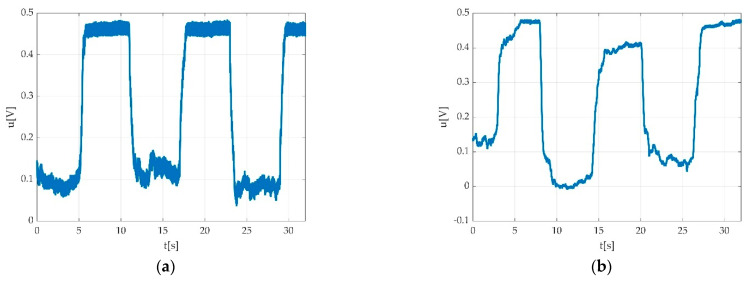
Examples of participant-derived errors; (**a**) good ability to maintain offset level; (**b**) offset unsteadiness due to inability of the participant to control the relaxation period; (**c**) unsteadiness of the offset during breaks between two MVICs; (**d**) inability to maintain the same level of force during MVIC, marked with a red circle.

**Figure 20 sensors-22-06310-f020:**
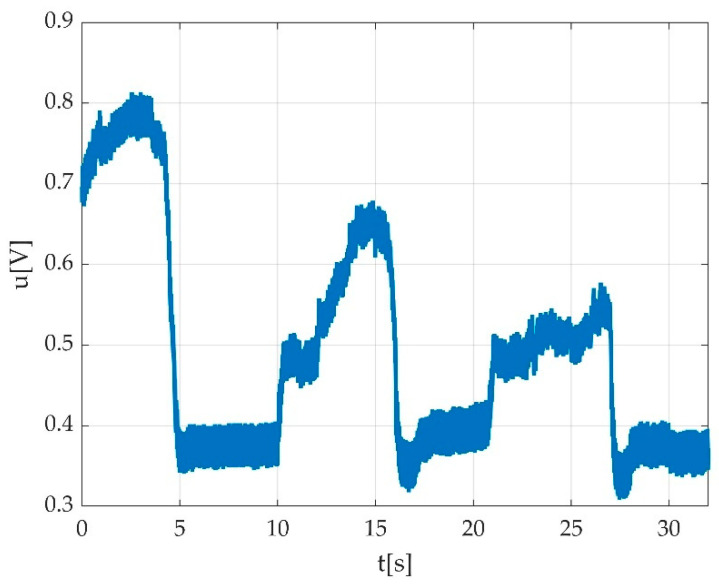
Example of invalid measurement due to participant inability to complete a maximal muscle effort during MVIC as per tester command.

**Figure 21 sensors-22-06310-f021:**
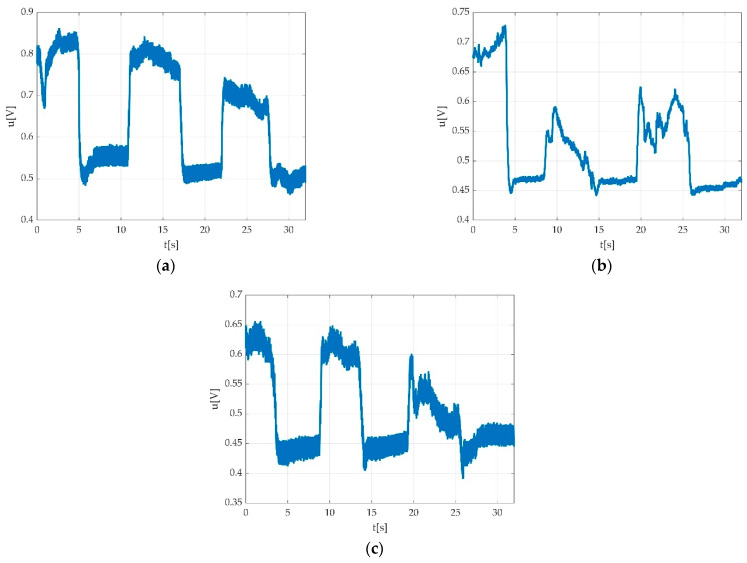
(**a**) Progressive decrement of voltage value due to participant fatigue while plantar flexing the foot; (**b**) invalid measurement due participant discomfort/pain derived from improper fixation of straps; (**c**) invalid measurement due to participant tremor during MVIC.

**Figure 22 sensors-22-06310-f022:**
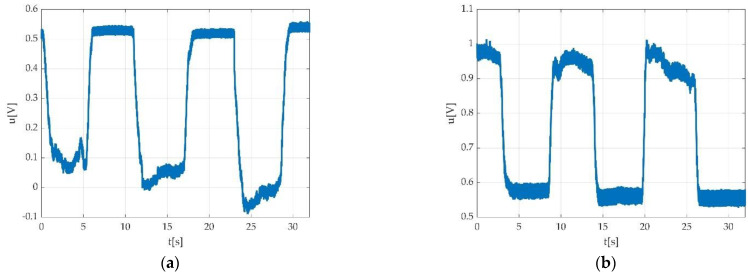
Example of not rejected time graphs: (**a**) contraction with the progressive increment of voltage value starting from first MVIC to the third MVIC; (**b**) muscle fatigue during the third MVIC during plantar flexion represented by decrement of voltage value; (**c**) progressive increment of voltage value during three consecutive MVICs (dorsiflexion); (**d**) progressive increment of voltage value during three consecutive MVICs (plantar flexion).

**Figure 23 sensors-22-06310-f023:**
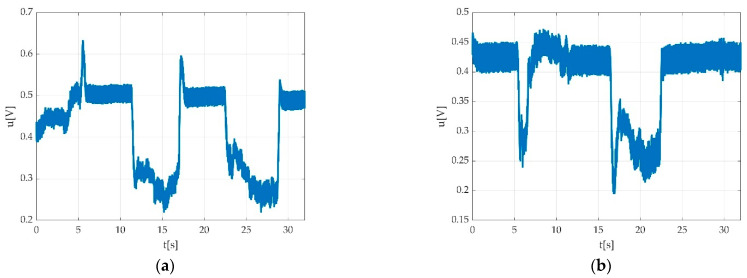
Example of invalid measurement due to: (**a**) participant inability to properly switch from effort to relaxation period during dorsiflexion; (**b**) participant inability to act according to the test protocol on tester command.

**Figure 24 sensors-22-06310-f024:**
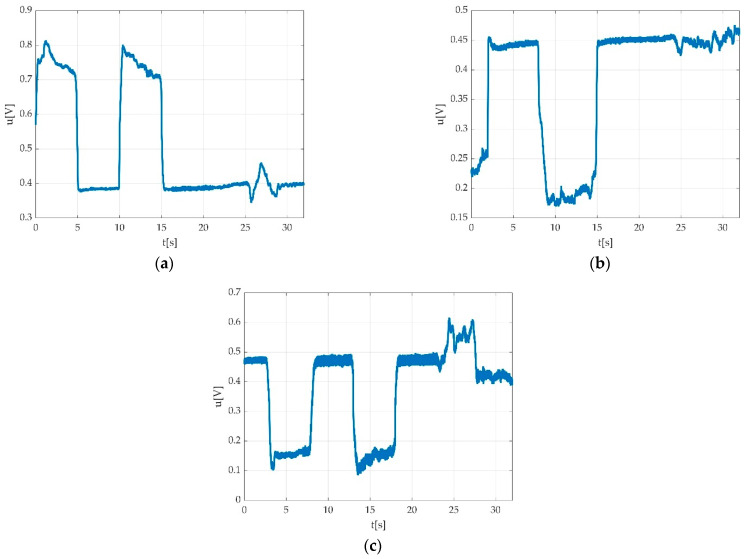
Example of invalid measurement due to improper tester command; (**a**) operator forgot to ask for third MVIC; (**b**) operator forgot to command relaxation; (**c**) while dorsiflexion command would have normally followed, the operator requested a plantar flexion command.

**Figure 25 sensors-22-06310-f025:**
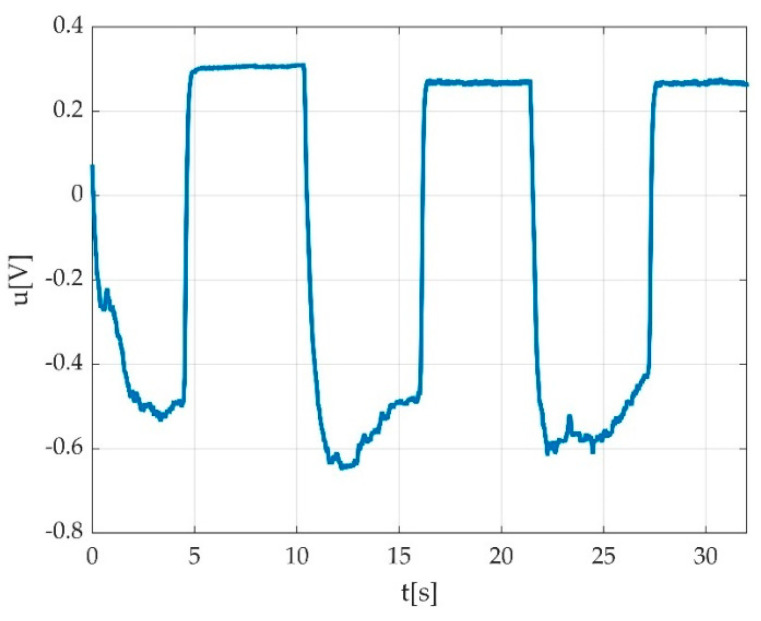
An example of dorsiflexion contraction indexed as a plantar flexion.

**Figure 26 sensors-22-06310-f026:**
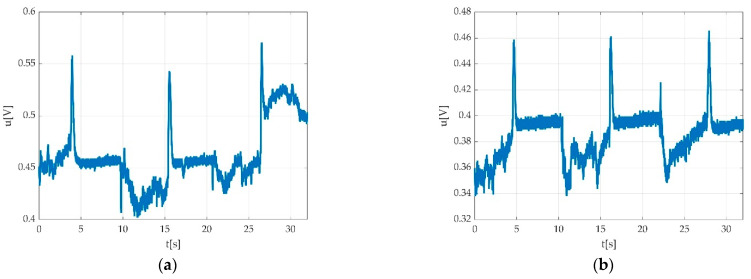
Example of invalid measurement rejected due to technical errors during (**a**) plantar flexion; (**b**) dorsiflexion.

**Figure 27 sensors-22-06310-f027:**
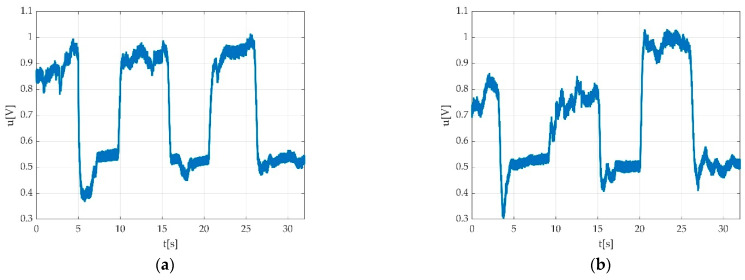
Example of invalid measurement due to: (**a**) impossibility to make automation detection of offset level; (**b**) participant inability to coordinate both muscle effort and relaxation period, which resulted in a time graph that does not permit an automatic detection of the offset level.

**Figure 28 sensors-22-06310-f028:**
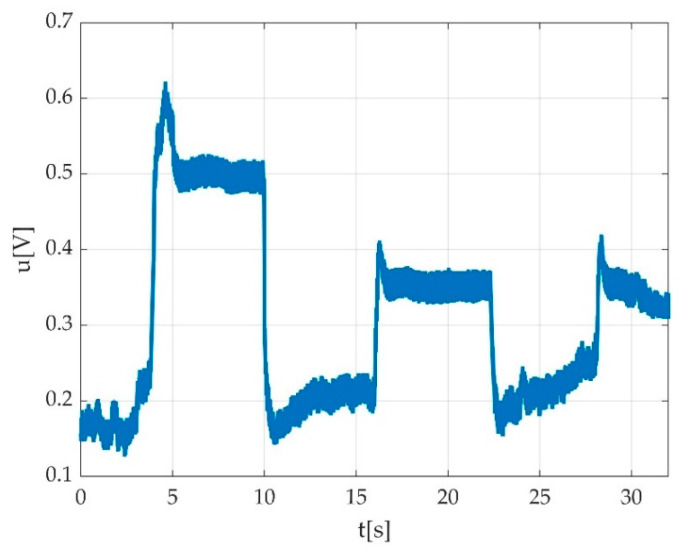
Example of automatically calculated offset steadiness types due to the inability to maintain constant offset level with progressive decrement of voltage from first to third MVIC due to muscle fatigue.

**Table 1 sensors-22-06310-t001:** Keywords in filenames for detection of the measurement characteristics from index file.

Keyword	Meaning
Sub XX	Identification number of the subject (participants), as digits in the XX place
LFT	Measurement is a flexion of the left foot
RX	Measurement is a flexion of the right foot
PFlex	Measurement is a plantar flexion
Dorsi	Measurement is a dorsiflexion
−5,	Initial pedal angle is −5° (comma is needed for detection)
+5,	Initial pedal angle is +5° (comma is needed for detection)
0,	Initial pedal angle is 0° (comma is needed for detection)
BASELINE	First measurement from multiple series
2h	Measurement at 2 h from the baseline in multiple series (similar for other time intervals—4 h, 6 h)

## Data Availability

Not applicable.

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
