# Peer review of "A Signal Processing Method for Assessing Ankle Torque with a Custom-Made Electronic Dynamometer in Participants Affected by Diabetic Peripheral Neuropathy"

_sensors, 2022, doi:10.3390/s22166310_

Round 1
Reviewer 1 Report
Thank you for providing me with the opportunity to review your manuscript.
The research deals with a complex topic, with great interest for the scientific community (especially for patients with diabetic peripheral neuropathy ).
This topic, maybe has an important clinical significance to population, but it´s not explain in the article enough. Perhaps it would be appropriate to include (or expand) this question.
I believe that there are some parts of the manuscript that can be improved to provide the scientific community with a better result of their research. (for example the clinical significance of the study results).
I provide the following comments which would strengthen the reporting of your work.
It is recommended to include some issues that are not taken into account: it is recommended to include more possible future lines of research at the end of the results, in the discussion (so far nonexistent) or before the conclusions.
In addition, it is necessary to include the ethical and legal part of the study in the body of the article and not in the final notes.It is recommended to include a separate epigraph for this in the "material and method" section.
I thank you for the trust placed in me by allowing me to review the article and I hope you can include the proposed suggestions RegardsAuthor Response
Dear Reviewer,
Thank you for your pertinent comments. We believe that we have covered all your requests.
The Authors

Reviewer 2 Report
The paper details a methodology that is often neglected or summarily presented in scientific papers, respectively an algorithm for removing aberrant values from a data set. Many of the studies fail or the conclusions are erroneous precisely because of the faulty way in which the scientific investigation is conducted. We appreciate this concern of the authors to generate an algorithm for signal processing used to assess ankle torque with a custom-made electronic dynamometer. Also, examples of invalid measurements may be of reference for future studies. We recommend some directions of improvement in terms of content and form of presentation of the work:
- -Materials and Methods. Lack of relevant information on the studies listed from line 129 to line 134. Who did these studies and under what conditions? Have the results of these studies been published? where? No bibliographical source is indicated. Who has the intellectual property rights for these studies? What protocols of research on human subjects have been followed? Who approved these protocols? Even if this information is synthetically presented at the end, it should be mentioned here as well.
- The explanations under each figure should be revised, avoiding the duplication of information and the clearer presentation of particular situations. For example: “Figure 23. Example of invalid measurement due to: a) participant inability to properly switch from effort to relaxation period during dorsiflexion; b) example of invalid measurement due to participant inability to act according to the testation protocol on testator command.” could be rephrased as: “Figure 23. Example of invalid measurement due to: a) participant inability to properly switch from effort to relaxation period during dorsiflexion; b) participant inability to act according to the testation protocol on testator command.”
Revise, similarly, the text for figures 24 and 27
- Replace the word “testation” with something else, maybe “testing” is much better: lines 71, 562, 613, 774, 784, 820.
- Define “maximal voluntary isometric contractions (MVIC)”. Maybe after line 99. Since there are several types of muscle contractions, not without relevance would be a justification: what makes this algorithm/equipment to be preferred for isometric contractions and no (or yes?) for other types? Reference to other bibliographic studies would be useful in this regard.
Author Response
Dear Reviewer,
Thank you for your pertinent comments. We believe that we have covered all requests.
The Authors
